# Pathogens of Medical Importance Identified in Hospital-Collected Cockroaches: A Systematic Review

**DOI:** 10.3390/microorganisms13020337

**Published:** 2025-02-04

**Authors:** Ariel Crespo, Yaxsier de Armas, Virginia Capó, Enrique Iglesias, Jaime Palomares-Marín, Luis Fonte, Arturo Plascencia-Hernández, Claudia L. Cueto-Aragón, Enrique J. Calderón, Héctor R. Pérez-Gómez

**Affiliations:** 1Centro Municipal de Higiene y Epidemiología, Bahia Honda 33800, Artemisa, Cuba; zeniamirandacuba@gmail.com; 2Pathology Department, Hospital Center, Institute of Tropical Medicine “Pedro Kourí”, Havana 11400, La Lisa, Cuba; yaxsier.dearmas@academicos.udg.mx (Y.d.A.); capovir@infomed.sld.cu (V.C.); 3Instituto de Patología Infecciosa y Experimental “Francisco Ruiz Sánchez”, Centro Universitario de Ciencias de la Salud, Universidad de Guadalajara, Guadalajara 44100, Jalisco, Mexico; 4Centro de Ingeniería Genética y Biotecnología, Havana 11400, La Lisa, Cuba; enrique.iglesias@cigb.edu.cu; 5Laboratorio de Ciencias Morfológico Forense y Medicina Molecular, Centro Universitario de Ciencias de la Salud, Universidad de Guadalajara, Guadalajara 44100, Jalisco, Mexico; jaime.palomares@academicos.udg.mx; 6Parasitology Department, Institute of Tropical Medicine “Pedro Kourí”, Havana 11400, La Lisa, Cuba; luisfonte@infomed.sld.cu; 7Departamento de Reproducción Humana y Crecimiento Infantil, Centro Universitario de Ciencias de la Salud, Universidad de Guadalajara, Guadalajara 44100, Jalisco, Mexico; aplascenciah@yahoo.com.mx; 8Hospital Infantil “Eva Sámano de López Mateos”, Morelia 58253, Michoacán, Mexico; claudiacueto22@gmail.com; 9Instituto de Biomedicina de Sevilla, Hospital Universitario Virgen del Rocío/Consejo Superior de Investigaciones Científicas/Universidad de Sevilla, 41013 Seville, Spain; ecalderon@us.es; 10Centro de Investigación Biomédica en Red de Epidemiología y Salud Pública (CIBERESP), 28029 Madrid, Spain

**Keywords:** cockroach, hospital, bacteria, virus, fungi, parasites, antibiotic resistance, healthcare-associated infections, pathogen transmission

## Abstract

Cockroaches serve as mechanical vectors for medically important pathogens, and their presence in hospitals is a common occurrence. This review summarizes the pathogens carried by cockroaches collected in hospitals around the world during the period 2000–2024 and focuses on their antibiotic resistance mechanisms and potential impact on the public health system. The conventional techniques are most used to identify microorganisms and determine antibiotic resistance, but there are few studies that use molecular techniques for bacterial identification and resistance mechanism detection. The species that appear most frequently in the selected articles were *Escherichia coli* (22 articles) and *Pseudomonas aeruginosa* (11 articles). Regarding antibiotic resistance, this review describes 79.0% (34/43) of the studies analyzed. *E. coli* and *P. aeruginosa* bacteria were found to be resistant to antibiotics in 51.2% and 25.6% of articles, respectively. The identification of pathogens carried by cockroaches collected in hospitals suggests a potential risk of these insects in the transmission of healthcare-associated infections, mainly in developing countries, where this issue is most alarming. The collected data suggest that integrated approaches to cockroach control and infestation management should be put in place based on scientific evidence.

## 1. Introduction

Cockroaches are ancient insects, with fossil evidence indicating that their lineage can be traced back to the Upper Carboniferous period [1]. Despite the passage of time, modern species have retained a high degree of morphological and physiological similarity to their ancestors. These insects possess the capacity to adapt to a diverse array of habitats, a trait that has enabled their survival in a multitude of environments [2,3].

The majority of these insects inhabit tropical and subtropical regions of the globe [1,4]. Approximately 4600 species have been identified, with over 99% classified as non-domestic [4]. Two notable examples of synanthropic cockroaches are *Blatella germanica* (Linnaeus, 1767) and *Periplaneta americana* (Linnaeus, 1758), which are among the most common due to their abundance and cosmopolitan distribution [1,5]. The species in question inhabits a variety of environments that provide optimal conditions for its survival, including residential buildings, commercial premises, healthcare facilities, educational institutions, and food-handling establishments. These insects are omnivorous, with a preference for foods rich in starch and sugars [4].

The World Health Organization (WHO) considers cockroaches a significant public health concern due to their role as mechanical vectors, capable of carrying potential pathogens on external and internal parts of their bodies. These pathogens include bacteria, fungi, protozoa, helminths, and viruses [1,6,7].

The contamination of food, surfaces, and objects of human use with pathogens carried by cockroaches occurs when these insects come into contact with them and regurgitate portions of their partially digested food or deposit their feces on them [3,4,8]. Pathogens from these insects can cause diseases such as bacterial dysentery, giardiasis, amebiasis, and toxoplasmosis, as well as symptoms such as nausea, abdominal pain, vomiting, and diarrhea [9,10]. Some papers show a high density of cockroaches in hospitals, which raises concerns about the potential adverse implications for the health of patients and healthcare workers [11,12,13].

In this regard, the WHO reports that healthcare-associated infections (HAIs) are responsible for approximately 40,000 deaths annually. It is estimated that 25% of HAIs occur in developing nations and between 5% and 15% in developed countries [7]. The predominant group of pathogens that cause these types of infections are bacteria, many of which are resistant to multiple antibiotics and have been isolated and identified from the bodies of cockroaches [14,15,16].

The phenomenon of antibiotic resistance (AR) represents a significant global public health concern, with documented associations with elevated mortality rates, prolonged hospitalizations, diminished therapeutic options, increased economic costs, and the potential for hospital-acquired outbreaks [7,17]. The gravity of this issue is underscored by projections indicating that by 2050, AR is likely to account for 10 million deaths and inflict substantial financial burdens, with a disproportionate impact on low- and middle-income countries [4].

The bacteria that most frequently cause infections in hospitalized patients include *S. aureus*, *Streptococcus* sp., *Acinetobacter* sp., *S. coagulase negative*, *P. aeruginosa*, *E. coli*, *Proteus mirabilis*, and *K. pneumoniae*. Some of these bacteria represent a global threat due to their increasing resistance to available antibiotics and their role in transmitting such resistance between bacteria of the same and different species [18,19,20].

Over the past decade, there has been a notable increase in research related to the pathogens carried by cockroaches collected in various environments, including hospitals [6]. However, there has been a paucity of studies that have synthesized this information with the aim of facilitating access to it and enhancing the understanding and comprehension of personnel working in such health institutions. Accordingly, the objective of the present study was to describe the pathogens carried by cockroaches collected in hospitals in different parts of the world during the period of 2000–2024 and their resistance mechanisms by means of a systematic review of the indexed literature.

## 2. Materials and Methods

### 2.1. Design

A systematic review of the literature was conducted to identify the pathogens and resistance mechanisms carried by cockroaches collected from hospitals in various parts of the world.

### 2.2. Methodology Used in the Search for Relevant Studies

The search strategy was conducted in accordance with the guidelines set forth in the PRISMA 2020 statement and through manual examination [21].

The following keywords were established: cockroach, hospital, bacteria, fungi, parasites, pathogen transmission, virus, hospital-acquired infection, and antibiotic resistance. In order to optimize the retrieval of relevant information, the bibliographic citations of the articles examined were selected.The literature search was conducted using Google Scholar and electronic databases pertinent to the field of health, namely LILACS, the SciELO portal, Scopus, Web of Science, and PubMed. The time interval was from January 2000 to December 2024.

### 2.3. Inclusion Criteria for Studies

Reports of isolation and identification of pathogenic organisms of medical importance from external or internal parts of cockroaches collected in hospital facilities.Articles that described susceptibility testing to antibiotics in bacteria.Articles published in the period from 1 January 2000 to 31 December 2024. For the establishment of this period of time, we took into account what was described by Guzman et al. [4] who express that since 2000 there has been an increase in publications in PubMed related to the topic of cockroaches and the pathogens identified from the bodies of these insects.Any scientific reports published in the form of a thesis, scientific article, or letter to the editor in English, Spanish, or Portuguese.

### 2.4. Exclusion Criteria for Studies

Articles in which the scientific name of the cockroach species collected is absent.Publications that constitute review articles.

### 2.5. Statistical Analysis

The GraphPad Prism v.10.1.2 software package (Boston, MA, USA) was used to present the results in figures.

## 3. Results

### 3.1. Search for Information

Figure 1 shows the flowchart of the selection process of the reports that make up the review based on the updated PRISMA guide [21]. Initially, using the established keywords, their combinations, date of publication, and languages, 1105 records on the subject were identified. 557 articles were excluded because they were considered duplicates. The 157 reports resulting from the screening of the records were examined based on the inclusion and exclusion criteria previously established by the authors. Finally, 43 articles were selected for review. Almost a third of the articles were published in PubMed (31.4%).

### 3.2. Distribution of Articles by Continent and Country

Appendix A shows that 93.0% of the research comprising the present review was conducted in developing nations. Of these, 58.1% were published in countries on the Asian continent, while 23.2% and 8.8% were in African and American nations, respectively. Iran (17), Brazil (3), Ethiopia (3), Iraq (3), Algeria (2), Nigeria (2), and Poland (2) accounted for 74.4% of the studies involved in this review (Figure 2).

### 3.3. Cockroach Species Collected and Identified in the Studies Analyzed

Six cockroach species were reported in the 43 studies analyzed [6,11,12,13,14,15,16,18,19,22,23,24,25,26,27,28,29,30,31,32,33,34,35,36,37,38,39,40,41,42,43,44,45,46,47,48,49,50,51,52,53,54]. *B. germanica* and *P. americana* were the most common, reported in 29 (64.4%) and 27 (62.7%) of the studies described, respectively. In 15 studies (34.8%), more than one cockroach species was captured in the same hospital environment (Figure 2).

### 3.4. Microorganisms Isolated and Identified from Cockroaches Collected in Hospitals

The bacteria isolated and identified from hospital-collected cockroach corpses are displayed in Appendix A. Bacteria represented the predominant group, being reported in 79% of the articles included in the review, followed by fungi (16.2%), helminths (11.6%), protozoa (9.3%), and viruses (6.9%), respectively.

Bacteria of potential medical importance appearing in the greatest number of articles were *E. coli* (22 articles), *P. aeruginosa* (11 articles), *S. aureus* and *Enterobacter cloacae* (10 articles), *Klebsiella oxytoca, K. pneumoniae,* and *Enterobacter aerogenes* (9 articles). *Citrobacter freundii* (7 articles) and *H. alvei* (6 articles), *Proteus vulgaris* and *S. marcescens* (5 articles). *Enterococcus faecalis*, *Morganella morganii, Proteus mirabilis*, *Staphylococcus epidermidis*, *Streptococcus pneumoniae*, *Streptococcus pyogenes,* and *Streptococcus agalactiae* appear less frequently (Appendix A).

Regarding fungi, three articles reported the identification of the species *Aspergillus niger* from the external part of cockroaches, while *Candida glabrata*, *Candida krusei,* and *Aspergillus fumigans* were isolated in a different study from those analyzed in the review. The remaining members of this group were not classified at the species level but at the genus level, such as *Mucor* spp., *Penicillium* spp., *Cladosporium* spp. and *Rhizopus* spp. (Appendix A).

Helminths harmful to human health were identified: *Enterobius vermicularis* (eggs, larvae, and adults) and *Ancylostoma duodenale* (larvae) in two and one of the studies analyzed, respectively. In addition, the presence of organisms belonging to the genera *Ascaris* (adults) and *Taenia* was detected (Appendix A).

Regarding protozoa, the presence of *Lophomonas blattarum* and *Entamoeba coli* was identified in three and one of the studies, respectively. In other studies, organisms such as *Blastocystis* spp., *Cyclospora* spp. and *Cystoisospora* spp. were reported (Appendix A).

Isolation and identification of medically important microorganisms from the external and internal parts of cockroaches was reported in 20 (46.5%) of the 43 articles reviewed. In all studies, pathogens were detected in both body parts of these vectors [6,14,15,22,23,25,27,29,31,32,33,34,35,39,42,46,49,50,52,54].

Three articles (8.5%) reported the identification of viruses (rotavirus and coronavirus) carried by cockroaches on the external and internal parts of their bodies [29,45,49].

Details on bacteria isolated/identified, as well as methods used to detect antibiotic resistance and its mechanisms in cockroaches, are stated in Appendix A in the Appendix A.

### 3.5. Antibiotic Resistance and Mechanisms of Resistance in Bacteria Identified in Cockroaches

Antibiotic resistance in bacteria isolated and identified from cockroaches collected in hospitals was addressed in 79.0% (34/43) of the studies included in this review.

Figure 3 shows that Gram-negative bacteria were identified in the highest percentage of the articles analyzed where AR is reported. In this regard, *E. coli* stands out with 22 studies (51.2%), followed by *P. aeruginosa* with 11 (25.6%), *E. cloacae* with 10 (23.3%). *K. oxytoca*, *K. pneumoniae,* and *E. aerogenes* are described in nine articles (20.9%). *C. freundii* is reported in seven studies (16.3%), *H. alvei* and *P. vulgaris* are reported in six (14.0%), and *S. marcescens* is reported in five (11.6%). In the case of Gram-positive bacteria, *S. aureus* appears in 10 studies (23.3%).

AR in bacteria was frequently observed in antibiotic groups such as penicillins, cephalosporins, aminoglycosides, amphenicols, tetracyclines, and quinolones [6,11,12,13,14,15,16,18,19,20,22,24,25,26,27,28,29,31,32,35,36,39,41,46,48,51,52].

Regarding antibiotic resistance mechanisms, the *BlaZ* gene coding for beta-lactamase enzymes and the production of extended-spectrum beta-lactamases were detected in two and nine studies, respectively [6,13,16,19,20,26,35,36,38,46,47]. These enzymes are responsible for the resistance mechanism characterized by inactivation of the antibiotic [55]. On the other hand, the presence of genes coding for penicillin-binding proteins involved in the resistance mechanism called target site modification [55] appears in one investigation [20]. Two articles reported the presence of *tet* (*K*) and *tet* (*L*) genes encoding proteins involved in the mechanism of antibiotic efflux through the energy-dependent pump. In addition, these studies were able to identify the *tet* (*M*) and *tet* (*O*) genes involved in the mechanism of target site alteration that protects the ribosome from the action of the antibiotic [6,20]. Finally, two additional studies in this review detected the mechanism of resistance (target site modification) to colistin mediated by the *mcr-1* gene in *E. coli* isolates and biofilm formation in the clonal lineage of methicillin-resistant *Staphylococcus aureus* isolates, respectively [46,54].

### 3.6. Methods for Identification of Microorganisms and Antibiotic Resistance

Appendix A shows that in 28 of the 34 studies (82.3%), bacteria were identified by seeding in culture media (general and differential), Gram staining (GS), and conventional biochemical tests (CBT). In addition, ten of the 34 articles referred to above (29.4%) used other techniques such as the Analytical Profile Index (API), serological tests, PCR, Vitek, and Matrix-Assisted Laser Desorption Ionization–Time of Flight Mass Spectrometry. In one of the 27 investigations, only the culture media seeding method and MALDI-TOF technology were used.

For the identification of fungi, methods based on seeding in culture media and examination of macroscopic and microscopic characteristics of the growths were used. Protozoa and helminths were classified using direct parasitological methods, and rotaviruses and coronaviruses were identified using ELISA or RT-qPCR technology (Appendix A).

As shown in Appendix A, the predominant method used to detect antimicrobial resistance was the use of disk diffusion (Kirby Bauer). This method was used in 28 of the 34 studies in which AR was investigated. On the other hand, PCR or RT-PCR was used in six of the 34 articles (17.6%), and the E-test method was used in one of the 34 studies (2.9%). It is worth noting that one of the studies reviewed did not report the method used to detect AR (Appendix A).

The detection of genes and enzymes that indicate the antibiotic resistance mechanism present in the bacteria analyzed was performed in 11 of the 34 studies (32.3%) [6,13,16,20,26,35,36,38,44,46,47]. The PCR or RT-PCR technique was used in eight of them, refs. [6,16,18,20,36,38,46,53]; in two, the Double Disk Synergy Test (DDST) [16,35]. In one was the performance E-test technique [36].

## 4. Discussion

A systematic review of pathogens of medical importance isolated from cockroaches collected in hospitals in different parts of the world over a 25-year period was performed. A literature search strategy similar to that used by other authors in reviews of isolated fly, cockroach, and beetle pathogens was established for the selection of studies [56,57]. This allowed a thorough process of identification, screening, and selection of articles based on the inclusion and exclusion criteria proposed by these authors.

Ninety-three percent of the studies in the review come from developing countries [58] in the Americas, Africa, and Asia [6,11,12,14,15,16,19,20,22,24,25,27,28,29,30,31,32,33,35,36,39,41,42,43,45,46,48,49,50,51]. Similar findings have been reported by different authors in their respective review articles [7,56,57]. In the countries where these studies were conducted, there are socioeconomic factors that negatively affect the health conditions found in second and third level hospital facilities [7,56,57,58]. The presence of cockroaches in hospitals and parasites in their integument and gastrointestinal tract is an indicator of poor sanitation in health facilities [1,15]. On the other hand, there are reports of cockroaches collected in hospitals in Spain, Japan, and France [2,7,18,37] showing that this phenomenon is independent of the geographical location and economic situation of nations, which becomes an epidemiological alert for developed countries.

*B. germanica* and *P. americana* were the predominant cockroach species in the studies included in this review [6,11,12,13,14,15,16,18,19,20,22,23,24,25,26,27,28,29,30,31,32,33,34,35,36,37,38,39,40,41,42,43,44,45,46,47,48,49,50,51,53,54]. Both are recognized as the most abundant of this group of insects worldwide [1,4,7]. Trade and the ability of both species to adapt to a wide range of conditions have played a key role in their distribution [1,2,4]. *Bl. orientalis*, *P. fuliginosa*, *P. japonica,* and *S. longipalpa* are other species considered to be of medical importance with limited worldwide distribution reported in the articles reviewed [1,2,18,48,52].

In fifteen of the 43 articles analyzed (34.8%), several species of cockroaches coexisting in the same hospital facility and carrying pathogens of medical importance were captured [6,11,18,19,20,22,30,31,41,46,48,49,52,53,54]. This fact could increase the likelihood of the spread and mechanical transmission of these organisms in the hospital environment. The high infestation of *P. americana* in the drainage system of hospitals, as well as its size, three to four times larger than *B. germanica*, could facilitate this phenomenon [6].

Cockroaches host and carry a variety of pathogens on the external and internal parts of their bodies, such as bacteria, infective forms of helminths, fungi, protozoa, and rotavirus, as reported by several authors [7,56,57]. The nocturnal habits of these vectors, their reproductive capacity, the ease with which they enter synanthropic places (homes, restaurants, hospitals, etc.), the ingestion of a wide variety of foods, including waste, and their free movement over a wide range of surfaces such as floors, tables, pipes, ceilings, cracks, and corridors condition the contact with ubiquitous microorganisms [1,4,37]. In addition, many pathogens can persist for months on dry inanimate surfaces, depending on the optimal conditions of temperature, humidity, and amount of inoculum, which facilitates the contact of cockroaches with them and consequently favors their mechanical vector capacity [1,4,15,59]. In this sense, different authors in their research do not find statistically significant differences between the distribution of potentially pathogenic bacteria identified on the outside and inside of the body of cockroaches captured in hospitals [6,18,22,26,27].

In fact, bacteria were the most frequently identified group in the studies included in the review. Several authors acknowledge the predominance of these microorganisms among the pathogens isolated from cockroaches [1,3,4,7]. This finding is not surprising given the high prevalence of these infectious agents in healthcare-associated infections [8,9]. However, Elgderi et al. [14] captured specimens of *P. americana* from hospitals and homes in Tripoli, Libya, showing no evidence that insects from hospitals are more likely to carry bacteria than those from homes. This suggests that the presence of these pathogens in these insects is primarily related to the sanitary conditions of the environments they inhabit.

Bacteria commonly identified in the review articles were *E. coli*, *S. aureus*, *E. aerogenes*, *K. pneumoniae*, *P. aeruginosa*, *C. freundii*, *E. cloacae*, *K. oxytoca*, and *P. vulgaris*, all of which are known to cause human infections, particularly in the hospital setting [60,61]. *K. pneumoniae* and *S. aureus* cause nosocomial pneumonia; *P. aeruginosa* causes surgical wound and burn infections; *E. coli* causes urinary tract infections; and both *E. aerogenes* and *E. cloacae* are opportunistic microorganisms that commonly cause infections in patients admitted to healthcare facilities [8,9,62]. These pathogens have been isolated from cockroaches collected from various parts of the hospital, such as the operating room, intensive care unit, and neonatal unit [4,7,12,15,24].

Several review articles suggest that cockroaches may play an important role in hospital-acquired parasitosis. Carzola et al. [33] identified enteroparasites of medical importance in *P. americana* cockroaches captured in Venezuela. Among the parasites identified were *E. vermicularis* and trophozoites of the protozoan *Lophomona blattarum*. It is important to highlight the fact that the same microorganisms were isolated from patients with pulmonary disease in Iran, China, Spain, India, and Peru [30,63]. In this regard, it is possible to speculate that nosocomial infection with these parasites might occur.

The studies analyzed in this research show the presence of fungi of medical importance carried by cockroaches collected in hospitals. Mycoses in hospital environments are common all over the world, mainly caused by *Aspergillus* spp. and *Candida* spp. [50,64]. In the same line, Motevalli-Haghi et al. [30] demonstrated the presence of different species of fungi, *A*. *flavus*, *A*. *niger*, *A*. *fumigatus*, *C*. *albicans,* and *C*. *glabrata,* in *B*. *germanica* and *P. americana* collected from three hospitals in Iran.

Viruses (coronaviruses, rotaviruses) were identified in only three studies in this review [29,45,49]. This finding is the first report of the identification of these infectious agents in cockroaches collected from hospitals in Ghana [29]. Rotavirus causes severe and fatal diarrhea in young patients worldwide and accounts for half of all hospitalizations for this condition in children under 5 years of age in developed countries. It is also responsible for approximately 25% of all hospital-acquired viral infections, particularly in immunocompromised children [65]. SARS-CoV-2 coronavirus was detected in two studies [45,49]. It is one of the main pathogens that primarily targets the human respiratory system; its high diffusion power, relatively high lethality, global spread, and lack of previous immunity in humans are the reasons for the importance of this disease [49].

However, the scarce detection of viruses in the articles included in this review may be due to the fact that the identification of pathogens in cockroaches tends to focus on bacteria, helminths, fungi, and protozoa due to the availability of resources and ease of diagnosis [1,29,65].

AR in bacteria is responsible for 70,000 patient deaths annually [55,62]. This phenomenon is reported in 14 articles in the current study, and mainly in bacteria of the ESKAPE group (*Enterobacter* spp., *E. coli*, *K. pneumoniae*, *S. aureus,* and *P. aeruginosa*) [61]. Moreover, other bacteria that are potentially harmful to humans have also been described: *H. alvei*, *P. vulgaris*, *S. marcescens*, *S. pneumoniae*, *S. agalactiae*, and *S. pyogenes*. These factors combine to create a global health risk associated with AR [62].

Bacterial resistance in cockroach species to a wide range of antibiotics such as penicillin, tetracycline, gentamicin, ceftaroline, aztreonam, chloramphenicol, cefepime, ceftazidime, erythromycin, ampicillin, and amickacin is highlighted [6,11,12,13,14,15,16,18,22,24,25,26,27,28,29,30,31,32,35]. Prado et al. [12] isolated 15 species of enterobacteria from *P. americana* captured in a public hospital in Brazil. Antibiotic susceptibility testing revealed resistance to most of the antimicrobials tested in the study. Abdolmalekiet et al. [6] provided the first report of phenotypic and genotypic evaluation of antibiotic resistance in Methicillin-Resistant *S. aureus* (MRSA) isolates from external and internal parts of *P. americana* and *B. germanica* collected in hospitals. The isolates also showed a high prevalence of resistance to the antibiotics penicillin, ceftaroline, tetracycline, gentamicin, and trimethoprim-sulfamethoxazole. In the neonatal ward of a hospital in the capital of Ethiopia, 400 cockroaches of the species *B. germanica* were collected. Of these, *K. pneumoniae*, *K. oxytoca*, *Providencia rettgeri*, *C. diversus*, *Citrobacter* spp., *E. cloacae*, *P. aeruginosa*, *E. coli*, *E. aeroginosa*, *S. aureus*, and *Acinetobacter* spp. were identified, and multidrug resistance was observed in all of them [35]. The above indicates that cockroaches may be involved in the spread of AR among bacteria [60].

Recently, Elizabeth M. Darby et al., in an excellent review on the molecular mechanisms of antibiotic resistance, summarized that a bacterial strain may develop multiple mechanisms of resistance to one or more antibiotics, and an antibiotic may be inactivated by different mechanisms in different bacterial species [55]. In the same way, various mechanisms of resistance are ‘intrinsic’, whereby the cell can use genes to survive antibiotic exposure, and some are ‘acquired’, whereby the gain of new genetic material provides new capacities that mediate survival [54]. Therefore, knowledge of these mechanisms is a fundamental step for the effective treatment of diseases with antimicrobial drugs, pivotal to recognizing global patterns of resistance and exploring how resistance genes contribute to the biology of the host [55]. In addition, understanding the molecular mechanisms developed by bacteria described new structural details of relevant resistance events, the identification of new resistance gene families, and the interactions between different resistance mechanisms.

In this review, the detection of AR mechanisms in bacteria was found in 14 of the 34 studies dealing with antibiotic resistance [6,13,16,19,20,26,34,35,36,38,44,46,47,53].

In Nigeria, using phenotypic methods, they found a mechanism of inactivation of beta-lactam drugs by the expression of extended-spectrum beta-lactamases (ESBL) [35]. Chehelgerdi et al. [20] described for the first time antibiotic resistance in Iran by phenotypic and genotypic pathways in *Streptococcus.* spp. isolated from cockroaches collected from hospital environments. These authors identified genes involved in the mechanisms of resistance to penicillins (pbp), tetracyclines (*tet K*, *tet M*, *tet O*, and *tet L*), macrolides (*erm* and *mef*), and streptogramins A and B (*rplV*) using PCR technique. In Tunisia, the production of BLEE (*blaCTX-M1*) and the presence of *mcr-1* genes responsible for the colistin resistance mechanism in the *E. coli* ST648 lineage were described for the first time [46].

In terms of the methods used to identify microorganisms and determine antibiotic susceptibility profiles in bacteria, conventional technologies such as seeding in culture media, Gram staining, conventional biochemical tests, and direct parasitological methods predominated. A possible explanation for this phenomenon is that these are the techniques used in developing countries, which were the basic scenarios in which the research was carried out. On the other hand, advanced technologies that provide better diagnostic quality and faster results (API [14,18,36], PCR [6,20,46], and MALDI-TOF [46]) were not available to most of the laboratories in the studies analyzed in this review. This fact affects the quality of the results, since in many cases the microorganism can only be identified at the genus level, which prevents us from knowing whether they are of medical importance. In addition, there are probably other pathogens that cannot be identified due to technological limitations.

On the other hand, in the detection of enzymes and genes that indicate the presence of antibiotic resistance mechanisms, it was observed that six articles included in this review employ the PCR or RT-PCR technique. However, in Japan, this technology was used for the same purpose in 2009 [18]. It is undeniable that these studies allow the identification of new pharmacological targets and the design of specific antibiotics to provide more precise treatments to combat infections caused by bacteria [55].

While some studies have identified cockroaches as vectors of multidrug-resistant bacteria in hospital settings, a definitive cause-and-effect relationship between cockroach exposure and patient infections remains to be established. Furthermore, only a limited number of studies have examined resistance mechanisms. However, the advent of molecular biology techniques has provided a framework for detecting genetic markers of antibiotic resistance and elucidating the transmission of pathogens from cockroaches to humans. It is imperative that future research address these points and that a worldwide distribution of molecular biology facilities and standardized protocols be implemented with priority given to underdeveloped countries. Public health systems on a global scale can benefit from this knowledge to track genetic variants of pathogens that may pose a threat due to their antibiotic resistance and epidemic or pandemic potential. Despite the superior sanitary conditions observed in developed countries, they are susceptible to the influence of various factors, including mass migration from developing countries and global trade that might facilitate the spread of cockroaches that harbor epidemiologically significant antibiotic-resistant organisms.

In light of the temporal and financial investments required for the development of novel antibiotic compounds to circumvent resistance mechanisms in microorganisms, it is imperative to focus on vector control (i.e., cockroaches) in the near future. In this regard, research lines must address the influence of environmental factors (e.g., temperature, humidity, and lighting) that trigger cockroach density and their impact on infestation in urban areas near health facilities and inside hospitals. It is also important to study the impact of sanitation measures (e.g., urban garbage removal, environmental cleanliness, and improvements in food distribution, handling, and storage conditions) and pest control practices. Concurrently, educational programs should be implemented to raise public and health personnel awareness through advertising on social media, training aids, and holding training courses. In essence, it is imperative to implement integrated approaches to cockroach control and infestation management based on scientific evidence.

The current research has the following limitation: in many studies, the nomenclature of the identified microorganisms was described down to the genus, which did not allow us to know the species and, consequently, its medical importance. For this reason, the analysis of the results emphasizes pathogens identified down to the species level.

## 5. Conclusions

Cockroaches in hospitals all over the world might carry potentially pathogenic organisms, which are recognized as etiological agents of human diseases belonging to the groups of bacteria, fungi, helminths, and protozoa. This reinforced the notion that the presence of these insects in healthcare facilities represents a potential risk due to their involvement in healthcare-associated infections and as reservoirs of multidrug-resistant bacteria. The impact of this phenomenon will be more alarming in developing countries, due to the excessive abuse of antibiotics and the fact that insect populations are rarely well controlled and have open access to many patient rooms in healthcare environments. Given the scarcity of publications on the subject addressed in this review, it is necessary to increase and deepen the study of pathogens of medical importance in cockroaches, a topic of importance for global health.

## Figures and Tables

**Figure 1 microorganisms-13-00337-f001:**
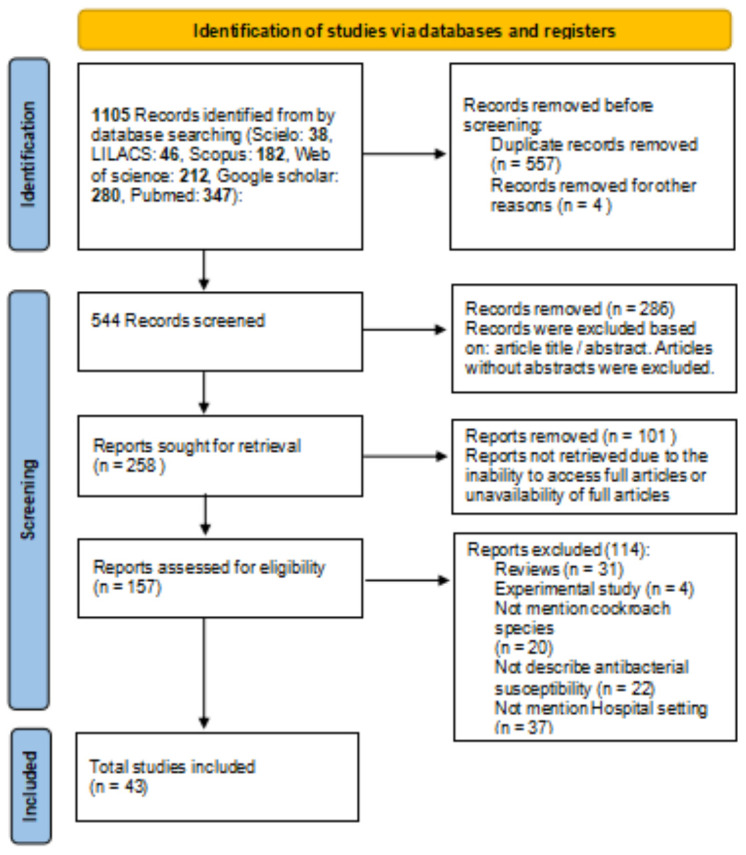
Flowchart of the report selection process based on the PRISMA guide.

**Figure 2 microorganisms-13-00337-f002:**
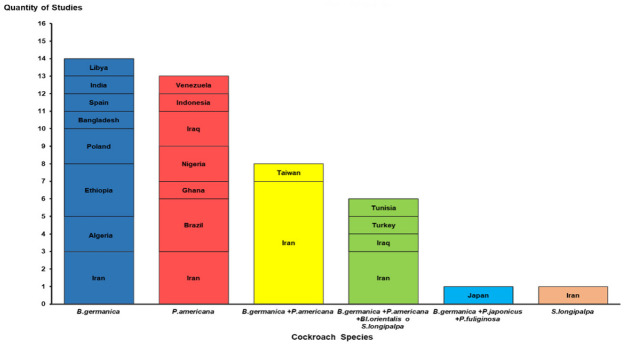
Distribution of studies by countries and species or group of cockroach species identified. *B: Blattella*; *P*: *Periplaneta*; *Bl*: *Blatta*; *S*: *Supella*.

**Figure 3 microorganisms-13-00337-f003:**
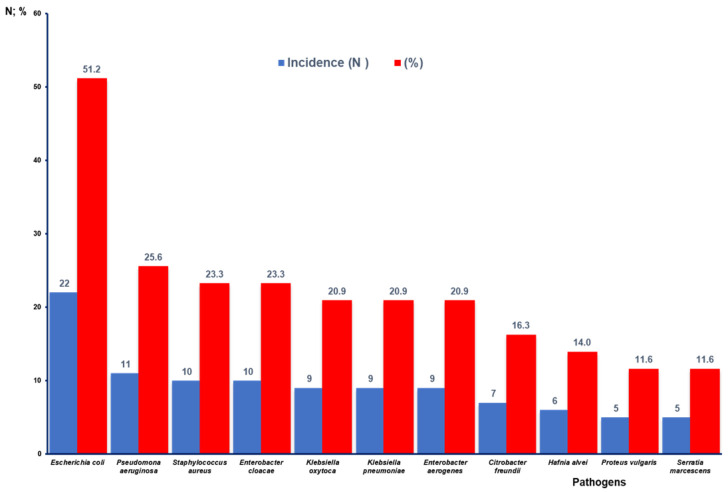
Number of publications and percentage according to reported bacteria with antibiotic resistance.

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
