# Peer review of "Pathogens of Medical Importance Identified in Hospital-Collected Cockroaches: A Systematic Review"

_microorganisms, 2025, doi:10.3390/microorganisms13020337_

Round 1
Reviewer 1 Report
Comments and Suggestions for Authors
After reviewing the provided manuscript titled "Pathogens of medical importance identified in hospital-collected cockroaches: a systematic review," here are some referee comments from the perspective of a reviewer:
1. The article provides a systematic review of medically important pathogens identified in hospital-collected cockroaches between 2000-2023. Please ensure that all data and references are up-to-date and reflect the current state of research.
2.The article should detail whether additional databases and search engines, such as Scopus, Web of Science, etc., should be included to ensure a comprehensive literature search.
3. The article notes that 95.4% of the studies are from developing countries. Please further discuss the potential impact of this geographical imbalance on the study's conclusions.
4. The article mentions antibiotic resistance, but only a few studies explore resistance mechanisms. Consider a more detailed discussion on this issue and its potential impact on public health.
5. The article uses Microsoft Office Excel 2007 to present results. Consider using more advanced statistical software for data analysis to improve accuracy and reliability.
6. Propose future research directions based on current study findings to advance the field.
Author Response
Referee 1
Comments and Suggestions for Authors
After reviewing the provided manuscript titled "Pathogens of medical importance identified in hospital-collected cockroaches: a systematic review," here are some referee comments from the perspective of a reviewer:
We appreciate the reviewer's comments on the knowledge that they will help improve the quality of the document. We provide our responses below each comment (written in red).
- The article provides a systematic review of medically important pathogens identified in hospital-collected cockroaches between 2000-2023. Please ensure that all data and references are up-to-date and reflect the current state of research.
We have carefully reviewed this aspect and all data and references are up-to-date and reflect the current state of research. With the news changes made to the document, the references have also been updated. In fact, we utilized 66 references in the new version.
- The article should detail whether additional databases and search engines, such as Scopus, Web of Science, etc., should be included to ensure a comprehensive literature search.
As the reviewer suggests, we included the Web of science (212 articles) and Scopus (182 article) databases with 212 and 182 articles, respectively. Please look at the PRISMA 2020 flow diagram for new systematic reviews which included searches of databases and registers.
- The article notes that 95.4% of the studies are from developing countries. Please further discuss the potential impact of this geographical imbalance on the study's conclusions.
After incorporating new articles into the review, 93.0% of the studies are from developing countries.
The potential causes and impact of such imbalance is discussed in the Discussion section of the manuscript where the following sentence was added: “Despite the superior sanitary conditions observed in developed countries, they are susceptible to the influence of various factors, including mass migration from developing countries and global trade that might facilitate the spread of cockroaches that harbor epidemiologically significant antibiotic-resistant organisms.”
We also discussed the potential impact of this geographical imbalance on the study's conclusions and added the following sentence: “The impact of this phenomenon will be more alarming in developing countries, due to the excessive abuse of antibiotics and that insect populations are rarely well controlled and have open access to many patient rooms in healthcare environments.” Also, in the abstract a similar idea was added.
- The article mentions antibiotic resistance, but only a few studies explore resistance mechanisms. Consider a more detailed discussion on this issue and its potential impact on public health.
Articles with new resistance mechanisms were added. Besides, in the Discussion section one paragraph was added (Lines 398-411):
“Recently, Elizabeth M. Darby et al., in an excellent review on the molecular mechanisms of antibiotic resistance, summarized that a bacterial strain may develop multiple mechanisms of resistance to one or more antibiotics, and an antibiotic may be inactivated by different mechanisms in different bacterial species. [56] In the same way, various mechanisms of resistance are ‘intrinsic’, whereby the cell can use genes to survive antibiotic exposure, and some are ‘acquired’, whereby gain of new genetic material provides new capacities that mediate survival.[55] Therefore, knowledge of these mechanisms is a fundamental step for the effective treatment of diseases with antimicrobial drugs, pivotal to recognizing global patterns of resistance and exploring how resistance genes contribute to the biology of the host. [56] In addition, understanding the molecular mechanisms that bacteria use to resist the action of antimicrobials described new structural details of relevant resistance events, the identification of new resistance gene families and the interactions between different resistance mechanisms.”
Another paragraph:
“While some studies have identified cockroaches as vectors of multidrug-resistant bacteria in hospital settings, a definitive cause-and-effect relationship between cock-roach exposure and patient infections remains to be established. Furthermore, only a limited number of studies have examined resistance mechanisms. However, the advent of molecular biology techniques has provided a framework for detecting genetic markers of antibiotic resistance and elucidating the transmission of pathogens from cockroaches to humans. It is imperative that future research address these points and that a worldwide distribution of molecular biology facilities and standardized protocols be implemented with priority given to underdeveloped countries. Public health systems on a global scale can benefit from this knowledge to track genetic variants of pathogens that may pose a threat due to their antibiotic resistance and epidemic or pandemic potential. Despite the superior sanitary conditions observed in developed countries, they are susceptible to the influence of various factors, including mass migration from devel-oping countries and global trade that might facilitate the spread of cockroaches that harbor epidemiologically significant antibiotic-resistant organisms.”
- The article uses Microsoft Office Excel 2007 to present results. Consider using more advanced statistical software for data analysis to improve accuracy and reliability.
In the new version of the manuscript, we are using the GraphPad Prism v.10.1.2 software package. This has been described in M&M section.
- Propose future research directions based on current study findings to advance the field
In the Conclusion section the following paragraph was added:
“In light of the temporal and financial investments required for the development of novel antibiotic compounds to circumvent resistance mechanisms in microorganisms, it is imperative to focus on vector control (i.e., cockroaches) in the near future. In this regard, research lines must address the influence of environmental factors (e.g., temperature, humidity, and lighting) that trigger cockroach density and their impact on infestation in urban areas near health facilities and inside hospitals. It is also important to study the impact of sanitation measures (e.g., urban garbage removal, environmental cleanliness, and improvements in food distribution, handling, and storage conditions) and pest control practices. Concurrently, educational programs should be implemented to raise public and health personnel awareness through advertising in social media, training aids, and holding training courses. In essence, it is imperative to implement integrated approaches to cockroach control and infestation management based on scientific evidence.”

Reviewer 2 Report
Comments and Suggestions for Authors
Abstract Review and Comment
The abstract provides a comprehensive overview of the systematic review, clearly outlining the study's objectives, methods, and key findings. The focus on cockroaches as mechanical vectors of medically significant pathogens in hospital environments is highly relevant, given their potential public health impact. The use of systematic review principles and adherence to PRISMA guidelines enhance the reliability and reproducibility of the study.
Strengths:
- Clarity of Objectives: The goal to describe pathogens and their resistance mechanisms carried by cockroaches in hospital settings from 2000-2023 is well-stated.
- Comprehensive Methodology: The inclusion of various databases and manual examinations strengthens the study’s validity.
- Key Findings Presented: The abstract effectively highlights the most commonly identified pathogens, resistance percentages, and methods employed for identification, providing valuable insights into the extent of the problem.
Areas for Improvement:
- Organization and Flow: The abstract could be more concise. Some sections, such as historical context or detailed descriptions of cockroach species, may be summarized or relocated to the introduction.
- Keywords: Consider including terms like "healthcare-associated infections (HAIs)" or "pathogen transmission" to improve searchability.
- Significance of Molecular Techniques: While the abstract notes a lack of molecular techniques, elaborating briefly on their importance could highlight the gap in existing research and future directions.
- Quantitative Details: Percentages or figures (e.g., 19% resistance in E. coli) are useful but would benefit from contextualization to underline the clinical implications.
Comment:
The abstract does a good job of summarizing the study but could be more concise and focused. Highlighting the significance of the findings in combating HAIs or informing hospital pest management policies would enhance its impact. Overall, it provides a solid foundation for further discussion of this critical issue.
Author Response
Referee 2
Comments and Suggestions for Authors
Abstract Review and Comment
The abstract provides a comprehensive overview of the systematic review, clearly outlining the study's objectives, methods, and key findings. The focus on cockroaches as mechanical vectors of medically significant pathogens in hospital environments is highly relevant, given their potential public health impact. The use of systematic review principles and adherence to PRISMA guidelines enhance the reliability and reproducibility of the study.
Strengths:
Clarity of Objectives: The goal to describe pathogens and their resistance mechanisms carried by cockroaches in hospital settings from 2000-2023 is well-stated.
Comprehensive Methodology: The inclusion of various databases and manual examinations strengthens the study’s validity.
Key Findings Presented: The abstract effectively highlights the most commonly identified pathogens, resistance percentages, and methods employed for identification, providing valuable insights into the extent of the problem.
We thank the reviewer for his comments and for recognizing the strengths of the work.
Areas for Improvement:
Organization and Flow: The abstract could be more concise. Some sections, such as historical context or detailed descriptions of cockroach species, may be summarized or relocated to the introduction.
The abstract has been refined and it was shortened eliminating superfluous data.
Keywords: Consider including terms like "healthcare-associated infections (HAIs)" or "pathogen transmission" to improve searchability.
Thanks for your comment. We have included those terms in the keywords.
Significance of Molecular Techniques: While the abstract notes a lack of molecular techniques, elaborating briefly on their importance could highlight the gap in existing research and future directions.
In the Conclusion section the following paragraph was added:
“While some studies have identified cockroaches as vectors of multidrug-resistant bacteria in hospital settings, a definitive cause-and-effect relationship between cock-roach exposure and patient infections remains to be established. Furthermore, only a limited number of studies have examined resistance mechanisms. However, the advent of molecular biology techniques has provided a framework for detecting genetic markers of antibiotic resistance and elucidating the transmission of pathogens from cockroaches to humans. It is imperative that future research address these points and that a worldwide distribution of molecular biology facilities and standardized proto-cols be implemented with priority given to underdeveloped countries. Public health systems on a global scale can benefit from this knowledge to track genetic variants of pathogens that may pose a threat due to their antibiotic resistance and epidemic or pandemic potential. Despite the superior sanitary conditions observed in developed countries, they are susceptible to the influence of various factors, including mass migration from developing countries and global trade that might facilitate the spread of cockroaches that harbor epidemiologically significant antibiotic-resistant organisms.”
Quantitative Details: Percentages or figures (e.g., 19% resistance in E. coli) are useful but would benefit from contextualization to underline the clinical implications.
In the new version, one table was added. Also, the clinical implications of principal bacteria were clarified.
Comment:
The abstract does a good job of summarizing the study but could be more concise and focused. Highlighting the significance of the findings in combating HAIs or informing hospital pest management policies would enhance its impact. Overall, it provides a solid foundation for further discussion of this critical issue.
In the new version of the Abstract these points were considered.

Reviewer 3 Report
Comments and Suggestions for Authors
The article with a promising title represents a review of publications studying the diversity of infectious pathogens spreading in hospitals by synanthropic insects (cockroaches of certain species), but its content is quite disappointing. So, half of the text is taken up by the description of the selection of the papers which, by the way, turned out to be only 22. Thus, the scientific soundness of the article is low, and I believe that it requires substantial modification.
Below I will give a few remarks that should be addressed both in the rebuttal letter and in the manuscript.
First of all, the Abstract needs to be refined, namely shortened and rephrased, in particular by removing the lines describing methods (31-35, 36-39, 42-45).
As for the whole text, the dots should be placed after the references in brackets, such as […]. instead of .[…]
Line 59: it seems like ‘two notable examples’ are given for non-domestic species.
Lines 68-69 and 98 are very similar to the Abstract fragments.
Lines 91-93 should be referenced.
The Methodology subsection should be shortened by removing of redundant lines 117-123, 125-126 and 132-136. By the way, it is not clear what limits the criterion from lines 132-134, and which exactly public places were taken into account in the article.
Why were 17 articles (while comparable n=22 were analyzed) that did not mention exact cockroach species (line 138) excluded if their analysis would also be useful for the review? In this case, the source of pathogens in Figure 2 could subsequently be displayed as “unspecified”.
The exclusion condition set out in lines 139-140 is not clear, as this would include laboratory analysis of pathogens spread by cockroaches in hospitals - i.e. the main methodology of the selected publications.
The article used a quite sparse list of keywords for the search. A great data massive was probably missed because the authors did not use the keywords “hospital-acquired infection”, “nosocomial infection”, "insect vectors", "insect-borne infection", “rotavirus”, etc. Publications found using these keywords should also be included in the study.
Another major omission of the authors is the absence of publications from 2024 among the analyzed material.
The inscriptions in the dotted rectangles in Figure 1 appear incomplete.
The correlation between the identified cockroach species (Figure 2) and the country of publication should be shown.
Lines 173-174 repeat almost verbatim the title of the subsection.
Table 1 is huge, so it needs to be moved to the Supplementary and only its short description should be leaved in the manuscript body (by the way, data on the pathogen incidence in cockroaches may be summarized in the table form). Moreover, the last column of Table 1 should be split to show separately the methods used for pathogen identification and AR analysis. In addition, here the country development class should be reflected and a column listing the AR species detected for each reference should be added. By the way, IM, PSA and MRA should be deciphered in the capture.
It is not necessary to describe in detail the methods by which the pathogen species and AR were detected in the reviewed articles, i.e. lines 224-245 should be shortened without separating them into subsections.
Lines 251-256 seem like a Conclusion fragment rather than a Discussion one.
The authors' suggestions for eliminating emerging infections caused by cockroach-distributed pathogens should be added to the Discussion, and other study limitations such as the small amount of data analyzed and the flaws of the methodology used for pathogen identification should be emphasized.
From lines 322-323, it is not clear whether Lophomona blattarum was isolated from the pulmonary patient or cockroach?
The reference list lacks DOIs and each reference is supplemented with unnecessary link to bibliographic database.
Author Response
Referee 3
Comments and Suggestions for Authors
The article with a promising title represents a review of publications studying the diversity of infectious pathogens spreading in hospitals by synanthropic insects (cockroaches of certain species), but its content is quite disappointing. So, half of the text is taken up by the description of the selection of the papers which, by the way, turned out to be only 22. Thus, the scientific soundness of the article is low, and I believe that it requires substantial modification.
In the new version of the manuscript papers from 2000-2024 were considered. Because of that the bibliography increased significantly.
Below I will give a few remarks that should be addressed both in the rebuttal letter and in the manuscript.
First of all, the Abstract needs to be refined, namely shortened and rephrased, in particular by removing the lines describing methods (31-35, 36-39, 42-45).
As suggested by the reviewer lines 31-39 and 42-45 were deleted from the Abstract. In addition, the Abstract was restructured and refined to provide a concessive summary of the review avoiding superfluous data.
As for the whole text, the dots should be placed after the references in brackets, such as […]. instead of .[…]
It was corrected in the new version of the manuscript.
Line 59: it seems like ‘two notable examples’ are given for non-domestic species.
It was corrected in the text. Now the sentence is “Two notable examples of synanthropic cockroaches are Blatella germanica (Linnaeus, 1767) and Periplaneta americana (Linnaeus, 1758) which are among the most common due to its abundance and cosmopolitan distribution.”
Lines 68-69 and 98 are very similar to the Abstract fragments.
In the new version of the manuscript the Abstract has changed and such similarities are lost.
Lines 91-93 should be referenced.
One reference was added
The Methodology subsection should be shortened by removing of redundant lines 117-123, 125-126 and 132-136. By the way, it is not clear what limits the criterion from lines 132-134, and which exactly public places were taken into account in the article.
Lines 117-123 were deleted and the time interval added to point 2.
Regarding Lines 125-126 we consider important as inclusion criteria and the phrase “collected in hospital facilities” was added (from lines 132-134). The previous is important to limit the search to areas inside hospitals not nearby to it.
Lines 132-134 were deleted.
We consider that lines 135-136 comprise an important inclusion criterion because it states that not all kinds of reports included in databases (like abstracts) were analyzed and also that not all languages were considered (for example Chinese or Russian were not included in the analysis). But we modified it as “Any scientific reports published in the form of a thesis, scientific article or letter to the editor in English, Spanish or Portuguese.”
Why were 17 articles (while comparable n=22 were analyzed) that did not mention exact cockroach species (line 138) excluded if their analysis would also be useful for the review? In this case, the source of pathogens in Figure 2 could subsequently be displayed as “unspecified”.
In the new version, following the PRISMA guide, this aspect has changed.
The exclusion condition set out in lines 139-140 is not clear, as this would include laboratory analysis of pathogens spread by cockroaches in hospitals - i.e. the main methodology of the selected publications.
Lines 139-140 were deleted because it is in fact an inclusion criterion already considered.
The article used a quite sparse list of keywords for the search. A great data massive was probably missed because the authors did not use the keywords “hospital-acquired infection”, “nosocomial infection”, "insect vectors", "insect-borne infection", “rotavirus”, etc. Publications found using these keywords should also be included in the study.
As suggested by the reviewer, in the new version we included other keywords. For that reason, a new PRISMA flowchart was created.
Another major omission of the authors is the absence of publications from 2024 among the analyzed material.
In the new version, publications from 2000 to 2024 were included.
The inscriptions in the dotted rectangles in Figure 1 appear incomplete.
It was corrected in the new version.
The correlation between the identified cockroach species (Figure 2) and the country of publication should be shown.
In the new version, the cockroach species and the country of publication are shown. Please, see Figure 2.
Lines 173-174 repeat almost verbatim the title of the subsection.
Lines 173-174 were rephrased as “The bacteria isolated and identified from hospital-collected cockroach corpses are displayed in Table 1.”
Table 1 is huge, so it needs to be moved to the Supplementary and only its short description should be leaved in the manuscript body (by the way, data on the pathogen incidence in cockroaches may be summarized in the table form). Moreover, the last column of Table 1 should be split to show separately the methods used for pathogen identification and AR analysis. In addition, here the country development class should be reflected and a column listing the AR species detected for each reference should be added. By the way, IM, PSA and MRA should be deciphered in the capture.
As suggested by the reviewer, table 1 was moved to the Supplementary data. Also, we created table 2 that showed the pathogen incidence in cockroaches. In fact, in table 1 one new column was created, it lists AR mechanisms in each study. The country and level of development was added to first column of Table 1
It is not necessary to describe in detail the methods by which the pathogen species and AR were detected in the reviewed articles, i.e. lines 224-245 should be shortened without separating them into subsections.
We partially agree with the reviewer’s comment and section 3.7 was added to 3.6 and the heading has changed accordantly. However, we consider that it is important for the readers to get the information about the techniques in use for pathogen identification and characterization of resistance mechanism. Techniques are just mentioned but protocols are not described.
Lines 251-256 seem like a Conclusion fragment rather than a Discussion one.
Unfortunately, we don't agree with the referee on this point. The paragraph in question is an introduction to the subject in the Discussion section, where the analysis and implications of the main paper's findings are continued. The manuscript already has a Conclusion section that reminds the reader of the context and purpose of the research. This section also summarizes the main findings, highlights their implications, and suggests areas for future research, as a conclusion should be.
The authors' suggestions for eliminating emerging infections caused by cockroach-distributed pathogens should be added to the Discussion, and other study limitations such as the small amount of data analyzed and the flaws of the methodology used for pathogen identification should be emphasized.
In the new version with PRISMA flowchart, two additional databases were analyzed and the flaws of the methodology used for pathogen identification were analyzed.
We included the idea of eliminating emerging infections caused by cockroach-distributed pathogens in the new version. See the last three paragraphs of the Discussion section.
From lines 322-323, it is not clear whether Lophomona blattarum was isolated from the pulmonary patient or cockroach?
We have rewritten the sentences to gain clarity. Now, in the new version of the paper “Among the parasites identified were E. vermicularis and trophozoites of the protozoan Lophomona blattarum. It is important to highlight the fact that the same microorganisms were isolated from patients with pulmonary disease in Iran, China, Spain, India, and Peru. [49, 68] In this regard, it is possible to speculate that nosocomial infection with these parasites might occur.
The reference list lacks DOIs and each reference is supplemented with unnecessary link to bibliographic database.
In the new version, DOIs were added and unnecessary link were deleted

Round 2
Reviewer 1 Report
Comments and Suggestions for Authors
Authors have revised the manuscript and it could be accepted.
Author Response
We appreciate the referee's review and comments. Thank you very much for accepting the article
Reviewer 3 Report
Comments and Suggestions for Authors
The authors refined the manuscript according to the reviewer's comments and significantly improved the text by adding required details or, on the contrary, by removing redundant fragments. The reference list was also updated by adding publications from recent years to the analysis. However, the figures remained quite blurred and their quality still needs to be improved. And Table 2 converted to a histogram would be more representative.
Author Response
Thanks for your comments.
As you suggested the figure was improved. Now the figure has higher quality.
As you suggested Table 2 was converted to a bar graph to make it more representative